# Robust and Faster Zeroth-Order Minimax Optimization: Complexity and Applications

**Weixin An[1], Yuanyuan Liu[1],\* Fanhua Shang[2],\* Hongying Liu[3,4]\***

[1]Key Laboratory of Intelligent Perception and Image Understanding of Ministry of Education,
School of Artificial Intelligence, Xidian University, China
[2]College of Intelligence and Computing, Tianjin University, China
[3]Medical School, Tianjin University, China
[4]Peng Cheng Lab, Shenzhen, China
weixinanut@163.com, yyliu@xidian.edu.cn, fhshang@tju.edu.cn,
hyliu2009@tju.edu.cn

## Abstract

Many zeroth-order (ZO) optimization algorithms have been developed to solve nonconvex minimax problems in machine learning and computer vision areas. However, existing ZO minimax algorithms have high complexity and rely on some strict restrictive conditions for ZO estimations. To address these issues, we design a new unified ZO gradient descent extragradient ascent (ZO-GDEGA) algorithm, which reduces the overall complexity to $\mathcal{O}(d\epsilon^{-6})$ to find an $\epsilon$-stationary point of the function $\psi$ for nonconvex-concave (NC-C) problems, where $d$ is the variable dimension. To the best of our knowledge, ZO-GDEGA is the first ZO algorithm with complexity guarantees to solve stochastic NC-C problems. Moreover, ZO-GDEGA requires weaker conditions on the ZO estimations and achieves more robust theoretical results. As a by-product, ZO-GDEGA has advantages on the condition number for the NC-strongly concave case. Experimentally, ZO-GDEGA can generate more effective poisoning attack data with an average accuracy reduction of 5%. The improved AUC performance also verifies the robustness of gradient estimations.

## 1 Introduction

We mainly consider a general regularized minimax problem:

$$\min_{x\in\mathbb{R}^{d_x}} \max_{y\in\mathbb{R}^{d_y}} \{\Psi(x,y) = g(x) + f(x,y) - h(y)\}, \tag{1}$$

where $f : \mathbb{R}^{d_x \times d_y} \to \mathbb{R}$ is nonconvex in $x$, concave in $y$ and $\ell$-smooth, $g : \mathbb{R}^{d_x} \to \mathbb{R}$ and $h : \mathbb{R}^{d_y} \to \mathbb{R}$ are convex but maybe nonsmooth functions. The problem (1) appears in many scenarios of machine learning, such as adversarial attack to regularized deep neural networks, regularized fair learning, and robust training [7, 13]. In this paper, we focus on the black-box setting of Problem (1), where the gradients are estimated by only functional values.

In the black-box setting, some evolutionary algorithms such as [46] have achieved good performance. However, they usually lack complexity analysis and may never converge to a solution due to pathological behavior [33]. Thus, ZO algorithms with convergence guarantees came into being. For example, ZO algorithms provide an alternative to higher-order optimization methods for solving robust network training with gradient or curvature regularization [37, 13]. Besides, ZO algorithms have also made considerable progress in escaping from saddle points [56].

---

\*Corresponding authors

38th Conference on Neural Information Processing Systems (NeurIPS 2024).

As for the black-box problem (1), ZO algorithms provide an access for solving it where the gradients are computationally infeasible or expensive, such as the Area Under Curve (AUC) maximization [54] with the ReLU activation and generating poisoning data to evaluate the black-box models [22, 33, 61]. In these scenarios, ZO algorithms become one of the best choices. For example, [33] proposed ZO-Min-Max for solving NC-strongly concave (NC-SC) data poisoning attack problems.

More practically, it is desired to solve Problem (1) under weaker conditions such as general concavity and more tolerant smoothing parameters of ZO estimators. But existing ZO algorithms such as [50] have high overall complexity for NC-C problems, and require picky smoothing parameters to ensure convergence, which weakens the robustness. On the other hand, the stochastic loss is actually more common [28]. But there is no ZO algorithm to solve the stochastic NC-C problem (1), which motivates us to fill this gap. Thus, **we summarize our motivations as follows:**

• Can we design a ZO algorithm to achieve a lower overall complexity with more weaker requirements on ZO estimators for NC-C problems?

• Due to the lack of research on stochastic ZO algorithms in the NC-C setting, can we develop a new stochastic ZO algorithm with theoretical guarantees?

Table 1: Comparison of the overall ZO oracle complexity of single-loop algorithms to find an $\epsilon$-stationary point of $f$ (Definition 3) or $\psi$ (Definition 2). $\kappa = \ell/\mu$ denotes the condition number, $d = d_x + d_y$ and $\widetilde{\mathcal{O}}(\cdot)$ hides logarithmic terms. Abbreviation: Settings (Set.), Algorithms (Algs.), Regular (Reg.), Theorem (The.).

| Set. | Algs. | $f/\psi$ | Reg.[1] | $\mu_1, \mu_2$[2] | Deterministic | Stochastic |
|------|-------|-----|------|------------|---------------|-----------|
| NC-C[3] | [50] | $f$ | $\mathcal{I}_{\mathcal{X}}, \mathcal{I}_{\mathcal{Y}}$ | $\mathcal{O}(d_x^{-1}\epsilon^2), \mathcal{O}(d_y^{-1}\epsilon^3)$ | $\mathcal{O}(d_x\epsilon^{-8}+d_y\epsilon^{-10})$ | Unknown |
| | [51] | $f$ | $\mathcal{I}_{\mathcal{X}}, \mathcal{I}_{\mathcal{Y}}$ | $\mathcal{O}(d_x^{-0.5}\epsilon^2), \mathcal{O}(d_y^{-0.5}\epsilon)$ | $\mathcal{O}(d\epsilon^{-4})$ | Unknown |
| | | $\psi$ | | $\mathcal{O}(d_x^{-1}\epsilon^4), \mathcal{O}(d_y^{-1}\epsilon^2)$ | $\mathcal{O}(d\epsilon^{-8})$[4] | Unknown |
| | **The. 2** | $\psi$ | $g, h$ | $\mathcal{O}(d_x^{-1}\epsilon), \mathcal{O}(d_y^{-1}\epsilon^2)$ | $\mathcal{O}(d\epsilon^{-6})$ | $\mathcal{O}(d_x\epsilon^{-8}+d_y\epsilon^{-10})$ |
| | **The. 3, 4** | $\psi$ | $g, h$ | $\mathcal{O}(d_x^{-0.5}\epsilon), \mathcal{O}(d_y^{-0.5}\epsilon)$ | $\mathcal{O}(d\epsilon^{-6})$ | $\mathcal{O}(d_x\epsilon^{-6}+d_y\epsilon^{-8})$ |
| NC-SC | [33] | $f$ | $\mathcal{I}_{\mathcal{X}}, \mathcal{I}_{\mathcal{Y}}$ | $\mathcal{O}(\kappa^{-3}d_x^{-1}\epsilon)), \mathcal{O}(\kappa^{-3}d_y^{-1}\epsilon)$ | — | $\mathcal{O}(\kappa^6 d\epsilon^{-6})$ |
| | [19] | $\psi$ | $\mathcal{I}_{\mathcal{X}}, \mathcal{I}_{\mathcal{Y}}$ | $\mathcal{O}(\frac{\epsilon^2}{d_x\sqrt{d\kappa^3}}), \mathcal{O}(\frac{\epsilon^2}{d_y d\kappa^3})$ | — | $\widetilde{\mathcal{O}}(\kappa^{4.5}d^{3/4}\epsilon^{-3})$ |
| | [47] | $\psi$ | $0, \mathcal{I}_{\mathcal{Y}}$ | $\mathcal{O}(\kappa^{-2}d_x^{-1.5}\epsilon), \mathcal{O}(\kappa^{-2}d_y^{-1.5}\epsilon)$ | $\mathcal{O}(\kappa^5 d\epsilon^{-2})$ | $\mathcal{O}(\kappa^5 d\epsilon^{-4})$ |
| | **The. 1** | $\psi$ | $g, h$ | $\mathcal{O}(d_x^{-1}\epsilon), \mathcal{O}(\kappa^{-0.5}d_y^{-1}\epsilon)$ | $\mathcal{O}(\kappa^2 d\epsilon^{-2})$ | $\mathcal{O}(\kappa^2(d_x+\kappa d_y)\epsilon^{-4})$ |

[1] About the column "Reg.", we transform Problem (1) into three models by setting different regularizers $g$ and $h$: (**i**) When $g$ and $h$ are both 0, Problem (1) is transformed into $\min_x \max_y \Psi(x,y) = f(x,y)$. (**ii**) When $g = \mathcal{I}_{\mathcal{X}}(\cdot)$ and $h = \mathcal{I}_{\mathcal{Y}}(\cdot)$, where $\mathcal{I}_C$ is the indicator function of the convex compact set $C$, Problem (1) is transformed into $\min_{x\in\mathcal{X}} \max_{y\in\mathcal{Y}} \Psi(x,y) = f(x,y)$. (**iii**) When $g = \mathcal{I}_{\mathcal{X}}(\cdot)$ and $h = \|y\|_1$, Problem (1) is transformed into $\min_{x\in\mathcal{X}} \max_y \Psi(x,y) = f(x,y) - \alpha\|y\|_1$.

[2] Larger smoothing parameters $\mu_1$ and $\mu_2$ imply that algorithms can tolerate rougher ZO estimations, which means stronger robustness. In some cases, robustness has a greater impact on algorithm performance than overall complexity, as shown in Table 2 below.

[3] For NC-C problems, [50, 51] depend on an extra monotonically decreasing regular parameter sequence and [50] also depends on Assumption 5. Our Theorem 2 does not depend on these assumptions.

[4] The work [51] only achieves an $\epsilon$-stationary point of $f$. According to [31, Proposition 4.12] and our analysis, [51] requires the overall complexity of $\mathcal{O}((d_x + d_y)\epsilon^{-8})$ to find an $\epsilon$-stationary point of $\psi$.

**Our contributions.** In this paper, we propose a unified algorithm to answer the above two questions. Our contributions can be summerized as follows.

• We design a unified single-loop ZO-GDEGA algorithm to solve the NC-C minimax problem (1) faster and more robustly. Specifically, we introduce the idea of continuous-time dynamics to assist in designing the update rules of dual variable $y$, which plays a key role in complexity analysis. Moreover, we analyze ZO-GDEGA by developing a concise theoretical framework, which reduces the overall complexity of finding a generalized $\epsilon$-stationary point of the function $\psi$ to $\mathcal{O}((d_x + d_y)\epsilon^{-6})$, even without the Lipschitz continuity assumption, as shown in Table 1.

• To the best of our knowledge, the stochastic ZO-GDEGA is the first ZO stochastic algorithm with theoretical guarantees for solving NC-C problems, which for the first time finds a generalized $\epsilon$-stationary point of $\psi$ with an overall complexity of $\mathcal{O}(d_x\epsilon^{-6} + d_y\epsilon^{-8})$, as shown in Table 1.

• Our ZO-GDEGA algorithm is more tolerant to the ZO estimations. Specifically, ZO-GDEGA allows a larger tolerance $\mathcal{O}(\epsilon)$ for smoothness parameters $\mu_1$ and $\mu_2$ compared with existing methods, as shown in Table 1, which enhances the robustness of ZO algorithms.

• As a by-product, ZO-GDEGA applied to NC-SC problems can obtain competitive complexity results $\mathcal{O}(\kappa^2(d_x + d_y)\epsilon^{-2})$ and $\mathcal{O}(\kappa^2(d_x + \kappa d_y)\epsilon^{-4})$ for deterministic and stochastic settings, respectively.

• Finally, the poisoning attack experiment shows that the poisoned data generated by ZO-GDEGA can reduce the accuracy by 2.1%-7.9% compared to baselines. The AUC maximization task shows that our algorithms can improve AUC performance by 0.4-5.4 units under rough ZO estimation.

## 2  Preliminaries and Related Work

This section provides several notations and definitions, and discusses some related works.

### 2.1  Notations

$\| \cdot \|$ and $\| \cdot \|_1$ denote the $\ell_2$-norm and $\ell_1$-norm of a vector, respectively. $\| \cdot \|_\infty$ denotes the $\ell_\infty$-norm of a vector, i.e., $\|x\|_\infty = \max_{1 \le i \le d_x} |x_i|$. We denote $a = \mathcal{O}(b)$ if $a \le Cb$ for some constant $C > 0$, any subgradient of $g(\cdot)$ at $x$ by $\partial g(x)$, a ZO estimator of $f(\cdot)$ at $x$ by $\hat{\nabla} f(x)$ for the gradient $\nabla f(x)$, the regularized coupling function by $\Gamma(x, y) := f(x, y) - h(y)$, the max function by $\Phi(x) := \max_y \{f(x, y) - h(y)\}$ and the regularized max function by $\psi(x) := g(x) + \Phi(x)$.

**Definition 1.** *The function $f(\cdot, \cdot)$ is $\ell$-smooth, i.e., $\|\nabla f(x, y) - \nabla f(x', y')\| \le \ell \|(x, y) - (x', y')\|$.*

### 2.2  Related Work

**ZO algorithms for nonconvex minimax optimization.** For solving NC-SC minimax problems, some single-loop ZO algorithms have been presented. For example, [33] proposed the ZO-Min-Max by integrating alternating stochastic Gradient Descent Ascent (GDA) method and ZO estimators, which finds an $\epsilon$-stationary point with the overall complexity $\mathcal{O}(\kappa^6 d\epsilon^{-6})$, when $g(x) = \mathcal{I}_{\mathcal{X}}(x)$ and $h(y) = \mathcal{I}_{\mathcal{Y}}(y)$. [47] proposed the ZO-GDA for solving the deterministic problem (1) with $g(x) \equiv 0$ and $h(y) = \mathcal{I}_{\mathcal{Y}}(y)$, and the ZO overall oracle complexity is bounded by $\mathcal{O}(\kappa^5 d\epsilon^{-2})$. Its stochastic variant, ZO-SGDA, can achieve the overall ZO oracle complexity $\mathcal{O}(\kappa^5 d\epsilon^{-4})$. Recently, [19] proposed an Acc-ZOMDA algorithm based on momentum acceleration techniques and further reduced the overall ZO oracle complexity. However, these algorithms have relatively strict restrictions on ZO estimators, as shown in Table 1, which affects their performance, as shown in Table 2.

As for NC-C problems, to the best of our knowledge, the only work proposed in [50] designs a ZO-AGP algorithm with the ZO oracle complexity $\mathcal{O}(d_x \epsilon^{-8} + d_y \epsilon^{-10})$, but it only considers the deterministic setting when $g(x) = \mathcal{I}_{\mathcal{X}}(x)$, $h(y) = \mathcal{I}_{\mathcal{Y}}(y)$, and $\mathcal{X}$ and $\mathcal{Y}$ are two convex compact sets, which limits its application to some scenarios such as regularized robust neural network training [21]. More recently, a new version of ZO-AGP [51] has been presented during the preparation of our work, which achieves lower complexity under the same constraints, but its theoretical results are still in terms of the coupling function $f$ instead of the stronger definition in terms of $\psi$, as shown in Table 1, while our analysis is in terms of $\psi$ and achieves lower complexity.

**Extragradient methods.** The extragradient (EG) method was first proposed in [27] for solving convex-concave saddle point problems and it adopts the gradient at the current point $x_t$ to find an intermediate point $x_{t+1/2}$ and then uses the gradient at $x_{t+1/2}$ to determine the next iteration point $x_{t+1}$. Specifically, at iteration $t$,

$$x_{t+1/2} = \text{prox}^g_{\eta_x}(x_t - \eta_x \nabla_x f(x_t)), \quad x_{t+1} = \text{prox}^g_{\eta_x}(x_t - \eta_x \nabla_x f(x_{t+1/2})), \qquad (2)$$

where $\text{prox}^g_\gamma(x) \triangleq \arg\min_z \{g(z) + \frac{1}{2\gamma}\|z - x\|^2\}$. On the one hand, the extra proximal gradient step guides the optimization process, which allows to escape cycling trajectories of the simultaneous gradient flow [23] and consider curvature information [40]. Compared with various GDA methods such as [31, 55], many works have shown the advantages of the first-order (FO) EG structure, such as handling noisy gradients in the convex-concave [26, 1] and NC-SC(C) [35] settings, but there is no insight into the ZO setting. On the other hand, in univariate convex optimization, the EG method has made considerable progress inspired by continuous time dynamic theory, and it can be explained as

an approximation of the more robust backward-Euler discretization [18], whereas gradient descent is a variant of the classical forward-Euler discretization [9]. Based on the two facts above, we aim to explore the performance of the EG structure in ZO minimax optimization to improve both theoretical and practical performance for solving the black-box problem (1).

**Lower bounds for minimax optimization.** For minimization problems, the lower bounds on ZO methods justify that the dependence on the dimension is inevitable without additional assumptions [12]. For minimax problems, will there be a similar conclusion? So far, researchers focus on the lower bounds of FO methods. For example, the lower bounds are $\Omega(\epsilon^{-1})$ [41] and $\widetilde{\Omega}(\kappa)$ [58] for convex-concave (C-C) and SC-SC settings, respectively. For NC-SC problems, there is also a lower bound for the FO setting [59, 30]. But to the best of our knowledge, there is no work proving a lower bound for ZO algorithms solving NC minimax problems. This paper focuses on another aspect and provides the upper bound for ZO algorithms solving NC minimax problems.

**Upper bounds for minimax optimization.** For the NC-C setting, GDA [31], Alternating GDA (AGDA) [4] and GDmax [25, 25] are analyzed to guarantee convergence. So far, the best complexity bound for FO deterministic NC-C problems is $\mathcal{O}(\epsilon^{-6})$. The works [60, 20] have further extended to the stochastic setting. For example, SAPD+ [60] achieved the complexity bound of $\mathcal{O}(\epsilon^{-6})$, which matches that of the deterministic case. For the NC-SC setting, the works [34, 49] proved the advanced bound of $\mathcal{O}(\kappa^3\epsilon^{-3})$. SAPD+ achieved another advanced complexity bound of $\mathcal{O}(\kappa\epsilon^{-4})$.

In minimizing optimization, the complexity of ZO methods is usually $d$ times that of corresponding FO methods [12]. In this sense, our results match the upper bound in the deterministic NC-C setting. Unfortunately, there is no similar conclusion in the minimax optimization yet. On the other hand, existing ZO methods such as [49, 50] focus on modifying existing FO algorithms, while this paper focuses on designing ZO minimax algorithms to directly improve performance.

## 3 ZO-GDEGA for NC-C Problems

In this section, we design a unified single-loop ZO gradient descent extragradient ascent (ZO-GDEGA) algorithm for solving the NC-C problem (1). The following assumptions are made for our analysis throughout this section.

**Assumption 1.** *We assume $f(x, \cdot)$ is concave for a given $x$.*

**Assumption 2.** *[4] The regularizers $g$ and $h$ are proper, convex and lower semicontinuous.*

*1. Additionally, $g$ is either $L_g$-Lipschitz continuous on its domain, which is assumed to be open, or the indicator of a nonempty, convex and closed set. Either of those assumptions guarantees the bound*

$$\|prox_\gamma^g(x) - x\| \leq \gamma L_g \tag{3}$$

*holds for any $\gamma > 0$ and all $x \in dom\, g$ (in the case of the indicator the statement is trivially true).*

*2. Furthermore, $h$ has a bounded domain $dom\, h$ such that the diameter of $dom\, h$ is bounded by $D_h$.*

Similar to [4], we also use the extension of the classical Danskin's theorem based on Assumptions 1 and 2 to guarantee that the solution set $Y^*(x) := \{y^*|y^* \in \arg\max_y\{f(x,y) - h(y)\}\}$ is non-empty for $\forall x \in \mathbb{R}^{d_x}$ and $\Phi(x)$ is $\ell$-weakly convex. Based on these facts, we propose and analyze our ZO-GDEGA algorithm for solving the NC-C problem (1) in deterministic and stochastic settings.

### 3.1 ZO-GDEGA in the Deterministic Setting

We first approximate the FO gradient by ZO randomized gradient estimators. Then, we propose the ZO-GDEGA algorithm to solve the black-box NC-C problem (1). Our algorithms are single-loop and more easily scalable compared to nested-loop algorithms such as [32].

***ZO randomized gradient estimators.*** We introduce the ZO randomized gradient estimators as follows: $\hat{\nabla}_x f(x,y) = \frac{1}{q_1}\sum_{i=1}^{q_1}\frac{d_x(f(x+\mu_1 u_i,y)-f(x,y))}{\mu_1}u_i, \hat{\nabla}_y f(x,y) = \frac{1}{q_2}\sum_{i=1}^{q_2}\frac{d_y(f(x,y+\mu_2 v_i)-f(x,y))}{\mu_2}v_i,$ where $\{u_i\}_{i=1}^{q_1} \subseteq \mathbb{R}^{d_x}$ and $\{v_i\}_{i=1}^{q_2} \subseteq \mathbb{R}^{d_y}$ are i.i.d. random direction vectors drawn uniformly from the unit Euclidean spheres, respectively. $\mu_1$ and $\mu_2$ are smoothing parameters, the conditions on which are strict in existing algorithms such as [51], while our algorithms allow them to be $\mathcal{O}(\epsilon)$, as shown in Table 1. The randomness caused by vectors $u_i$ and $v_i$ is coupled in the alternating

updates of $x$ and $y$ and undoubtedly increases the difficulty of complexity analysis. We introduce two smooth functions $f_{\mu_1}(x, y) = \mathbb{E}_u[f(x + \mu_1 u, y)]$ and $f_{\mu_2}(x, y) = \mathbb{E}_v[f(x, y + \mu_2 v)]$ to bridge the ZO estimators and the FO gradient, thereby assisting in complexity analysis as shown in Fig. 1.

---

**Algorithm 1** Deterministic Zeroth-Order Gradient Descent Extragradient Ascent Algorithm

---

**Initialize:** $x_0$, $z_0 = y_0$, step sizes $\eta_x$ and $\eta_y$.
  1: **for** $t = 0, 1, \ldots, T - 1$ **do**

  2:    $\hat{\nabla}_x F(x_t) = \begin{cases} \hat{\nabla}_x f(x_t, z_t) \text{ for NC-C case;} \\ \hat{\nabla}_x f(x_t, y_t) \text{ for NC-SC case;} \end{cases}$;    $x_{t+1} = \text{prox}_{\eta_x}^g (x_t - \eta_x \hat{\nabla}_x F(x_t))$;

  3:    $z_{t+1} = \text{prox}_{\eta_y}^h (y_t + \eta_y \hat{\nabla}_y f(x_t, y_t))$;        $y_{t+1} = \text{prox}_{\eta_y}^h (y_t + \eta_y \hat{\nabla}_y f(x_{t+1}, z_{t+1}))$;

  4: **end for**
  5: Randomly draw $\hat{x}$ from $x_1, \ldots, x_T$ at uniform.
**Output:** $\hat{x}$.

---

To improve performance, we integrate the EG structure and the ZO randomized gradient estimators, and propose the ZO-GDEGA algorithm. Specifically, it contains a proximal gradient descent step and two proximal gradient ascent steps, as shown in Algorithm 1.

### 3.1.1 Update rule of $x$.

Algorithm 1 updates $x$ by minimizing the following linearized approximation,

$$x_{t+1} = \arg \min_x \{g(x) + \langle x - x_t, \hat{\nabla}_x f(x_t, z_t)\rangle + \tfrac{1}{2\eta_x}\|x - x_t\|^2\}, \tag{4}$$

where $z_t$ is an auxiliary variable and its update rule is given later. Our design for updating $x_{t+1}$ relies only on the previous point $x_t$, and does not require $x_{t+1/2}$ as in the standard EG method (2).

### 3.1.2 EG update rule of $y$.

The quality of the solution set $Y^*(x)$ plays a key role on complexity. To take full advantage of the EG structure based on the concavity w.r.t. $y$, we integrate it into the update of $y$, which guides a new high-level idea from the continuous-time dynamic perspective.

*A high-level idea.* Here we show an intuition for the advantages of our algorithm. If $h$ is "simple", in the sense that the proximal operator has a closed-form solution [6]. Without loss of generality, we consider the case of $h(y) \equiv 0$. In this case, the proximal operator becomes the identity transformation. In fact, one-step ZO estimation w.r.t. $y$ can be viewed as the discretization of the stochastic continuous-time dynamic $\dot{Y}(t) = \hat{\nabla}_y f(X, Y(t))$ with the limit $\eta_y \to 0$. To get more robust and accurate solutions, we use the backward-Euler instead of forward-Euler discretization to solve this dynamic. An exact backward-Euler step implies the discretization $y_{t+1} = y_t + \eta_y \hat{\nabla}_y f(x_{t+1}, y_{t+1})$, but obtaining $y_{t+1}$ in such a manner could be computationally prohibitive. Instead, we opt for an extra gradient ascent step $z_{t+1} = y_t + \eta_y \hat{\nabla}_y f(x_t, y_t)$ to first approximate $y_{t+1}$ once as shown in the $3^{rd}$ line of Algorithm 1, which can be viewed as a "trial" step. Then, we use the ZO oracle at point $(x_{t+1}, z_{t+1})$ to approximate one-step backward-Euler discretization as shown in the end of the $3^{rd}$ line, which is one of the reasons why our ZO-GDEGA algorithm is more robust than [51].

### 3.1.3 Advantages over Existing Methods.

Compared with existing works, our ZO-GDEGA algorithm has the following three main advantages:

• *Stronger robustness.* About ZO estimators, ZO-GDEGA can tolerate smoothness parameters $\mu_1$ and $\mu_2$ of size $\mathcal{O}(\epsilon)$, whereas existing ZO algorithms have stricter restrictions on smoothness parameters such as $\mathcal{O}(\epsilon^4)$ as shown in Table 1, which is the fundamental reason why ZO-GDEGA improves the robustness. Specific proofs and experimental verifications are in subsequent sections.

• *Lower per-iteration complexity.* Compared with the standard EG method [35], Algorithm 1 only uses the EG structure for the update of $y$ instead of both $x$ and $y$, which reduces per-iteration complexity and maintains the same iteration complexity as the standard EG method. In fact, our ZO-GDEGA algorithm is easily extended to an FO method, which still reduces the computation cost and maintains the corresponding theoretical result. Since this paper mainly focuses on the ZO case, the FO variant of Algorithm 1 is shown in the Appendix.

• **More extensive applications.** The proximal operators generalize the projection operators in existing works such as [51]. Compared with [21], our ZO-GDEGA does not require additional compactness of the domains, which significantly extend the applicability.

## 3.2 ZO-GDEGA in the Stochastic Setting

We consider that $f$ is a stochastic function on a distribution $\mathcal{D}$, i.e., $f(x,y) := \mathbb{E}_{\xi \sim \mathcal{D}} f(x,y;\xi)$, and we can only access the stochastic function values $f(x,y;\xi)$. In this case, we introduce the stochastic version of ZO estimators: $\hat{\nabla}_x f(x,y;\mathcal{I}_1) = \frac{1}{b_1} \sum_{j=1}^{b_1} \hat{\nabla}_x f(x,y;\zeta_j) = \frac{1}{b_1} \sum_{j=1}^{b_1} \frac{f(x+\mu_1 u_j, y;\zeta_j) - f(x,y;\zeta_j)}{\mu_1/d_x} u_j$ as well as $\hat{\nabla}_y f(x,y;\mathcal{I}_2)$, where $\mathcal{I}_1 = \{\zeta_j\}_{j=1}^{b_1}$, $\mathcal{I}_2 = \{\xi_j\}_{j=1}^{b_2}$ denote mini-batch sets of $b_1$ and $b_2$ i.i.d. samples. According to [19], $\mathbb{E}_{U,\mathcal{I}_1}[\hat{\nabla}_x f(x,y;\mathcal{I}_1)] = \nabla_x f_{\mu_1}(x,y)$ and $\mathbb{E}_{V,\mathcal{I}_2}[\hat{\nabla}_y f(x,y;\mathcal{I}_2)] = \nabla_y f_{\mu_2}(x,y)$ with $U = \{u_i\}_{i=1}^{b_1}$ and $V = \{v_i\}_{i=1}^{b_2}$, and we need the common Assumptions 3, 6 and 7 in ZO stochastic optimization. Due to the page limit, we place Assumptions 6 and 7 in the Appendix.

**Assumption 3.** *The variance of the ZO stochastic estimators are bounded, i.e.,* $\mathbb{E}_{u,\zeta} \|\hat{\nabla}_x f(x,y;\zeta) - \nabla_x f_{\mu_1}(x,y)\|^2 \leq \sigma_1^2$, $\mathbb{E}_{v,\xi} \|\hat{\nabla}_y f(x,y;\xi) - \nabla_y f_{\mu_2}(x,y)\|^2 \leq \sigma_1^2$.

Based on the above analysis and Algorithm 1, we design a stochastic variant of ZO-GDEGA, as shown in Algorithm 2 in the Appendix. The main difference between Algorithms 2 and 1 is that the ZO estimators are replaced by their stochastic versions. In fact, for NC-C minimax problems, stochastic ZO-GDEGA is the first stochastic ZO algorithm and we prove its complexity in Section 5.

# 4 ZO-GDEGA for NC-SC Problems

In this section, we provide two by-products of our ZO-GDEGA algorithm for the NC-SC setting, as also shown in Algorithms 1 and 2. We design and analyze them based on Assumption 4.

**Assumption 4.** *We assume* $f(x,\cdot)$ *is* $\mu$-*strongly concave in* $y$ *for a given* $x$. *Moreover, the regularizers* $g$ *and* $h$ *are proper, convex and lower semicontinuous.*

Based on Assumption 4, we know that the max function $\Phi(x)$ is $(\kappa+1)\ell$-smooth, where $\kappa = \ell/\mu \geq 1$ is the condition number, which plays a key role in our analysis. Note that there are two main differences between our ZO-GDEGA for solving NC-SC and NC-C problems as follows:

• Due to the strong concavity, there is no need to use the intermediate point $z_t$ to update $x_{t+1}$, but directly use the ZO estimators at the point $(x_t, y_t)$ to accelerate updating, which contributes to achieve competitive complexity under weaker restrictions on $\mu_1$ and $\mu_2$, thereby enhancing robustness.

• The solution set $Y^*(x)$ is a singleton, which consists of a single element $y^*(x)$, thus we can use the quantity $\delta_t := \|y_t - y^*(x_t)\|^2$ to measure the quality of proximal ZO extragradient ascent steps.

In summary, we propose a unified ZO algorithm for solving black-box NC minimax problems, which outperforms existing ZO algorithms. Next, we rigorously analyze its complexity.

# 5 Theoretical Analysis

In this section, we first define two types of *generalized $\epsilon$-stationary points* as termination criteria. Then, we provide the complexity results of our ZO-GDEGA to find an $\epsilon$-stationary point.

## 5.1 Termination Criteria

We generalize two classical definitions of $\epsilon$-*stationary point* in [31] and [32] as follows.

**Definition 2.** *In the NC-C setting, a point $x$ is an $\epsilon$-stationary point of an $\ell$-weakly convex function $\psi$ if its Moreau envelope[2] $\psi_{1/2\ell}(x)$ satisfies $\|\nabla \psi_{1/2\ell}(x)\| \leq \epsilon$; In the NC-SC setting, a point $x$ is an $\epsilon$-stationary point of a function $\psi$ if $dist(0, \partial \psi(x)) \leq \epsilon$.*

**Definition 3.** *A pair of points $(x,y)$ is an $\epsilon$-stationary point of the coupling function $f$ if we have* $\|\mathcal{G}(x,y)\| \leq \epsilon$, *where* $\mathcal{G}(x,y) = \begin{bmatrix} \ell[x - prox_{1/\ell}^g(x - (1/\ell)\nabla_x f(x,y))] \\ \ell[y - prox_{1/\ell}^h(y + (1/\ell)\nabla_y f(x,y))] \end{bmatrix}$.

---

[2] $\psi_{1/2\ell}(x)$ is the Moreau envelope of $\psi(x)$ if $\psi_{1/2\ell}(x) = \min_w \{\psi(w) + \ell \|w - x\|^2\}$ for each $x \in \mathbb{R}^{d_x}$.

Based on [31, Proposition 4.11 and 4.12], the following two propositions clarify the relationship between the generalized Definitions 2 and 3.

**Proposition 1.** *Under Assumptions 1 and 2, if a point $(\hat{x}, \hat{y})$ is an $\epsilon^2/(\ell D_h)$-stationary point in terms of Definition 3, a point $\hat{x}$ is an $\mathcal{O}(\epsilon)$-stationary point in terms of Definition 2.*

**Proposition 2.** *Under Assumption 4, if a point $(\hat{x}, \hat{y})$ is an $\epsilon/\kappa$-stationary point in terms of Definition 3, a point $\hat{x}$ is an $\mathcal{O}(\epsilon)$-stationary point in terms of Definition 2.*

Thus, we can draw a similar conclusion as in [31]: The $\epsilon$-stationary point definition of $\psi$ is stronger than that of $f$. To obtain tighter complexity bounds, we analyze ZO algorithms for the first time based on $\epsilon$-stationary point definition of $\psi$ (i.e., Definition 2).

## 5.2 Complexity Analysis for NC-SC Problems

We first analyze the complexity of ZO-GDEGA for the NC-SC setting in deterministic and stochastic cases. Our analysis is a non-trivial extension of the proofs in [31] due to the following challenges.

***Key technical challenges.*** In this case, bounding $\sum_{t=0}^{T} \delta_t$ is a key step. But the proximal operators and the ZO EG structure make it much more difficult to bound this term than existing works. To address this challenge, we propose to establish the upper bound on $\sum_{t=0}^{T} \delta_t$ in terms of $\sum_{i=0}^{T} \|x_i - x_{i+1}\|^2$ (see the Appendix for details), ***which also reduces the overall complexity's dependence on $\kappa$, and results in the coefficients of $\mu_1^2$ and $\mu_2^2$ being only $\mathcal{O}(1)$ and $\mathcal{O}(\kappa)$, thereby weakening their dependence on $\kappa$ and enhancing robustness.*** Here we directly give the complexity results.

**Theorem 1.** *We choose stepsizes $\eta_x \leq \frac{1}{256\kappa^2\ell}$, $\eta_y = \frac{1}{2\ell}$, $\mu_1 = \mathcal{O}(d_x^{-1}\epsilon)$ and $\mu_2 = \mathcal{O}(\kappa^{-1/2}d_y^{-1}\epsilon)$. Under Assumption 4, our ZO-GDEGA can find an $\epsilon$-stationary point in terms of Definition 2, i.e., $\min_{1 \leq t \leq T} dist(-\partial g(x_t), \nabla\Phi(x_t)) \leq \epsilon$, with the overall ZO oracle complexity $\mathcal{O}(\kappa^2(d_x + d_y)\epsilon^{-2})$ for deterministic and $\mathcal{O}(\kappa^2(d_x + \kappa d_y)\epsilon^{-4})$ for stochastic settings.*

Theorem 1 shows that ZO-GDEGA allows for larger $\mu_1$ and $\mu_2$ than compared algorithms (see Table 1), which means that our algorithms can tolerate rougher ZO estimations. Besides, our complexity bounds have the weaker $\kappa$ dependence than them. Thus, even our by-products have some advantages. Moreover, this analysis also builds a bridge for analyzing the NC-C setting.

## 5.3 Complexity Analysis for NC-C Problems

In this part, we analyze ZO-GDEGA in the NC-C setting. We provides two perspectives: continuity-agnostic (more relaxed condition) and continuity-dependent (better bound) analysis.

### 5.3.1 Continuity-Agnostic Complexity Analysis.

We first analyze the complexity for the NC-C setting based on Theorem 1 and smooth technology in [38]. By adding a smoothing term, we approximate the NC-C problem (1) with the following NC-SC model: $\min_x \max_y \{\hat{\Psi}(x, y) = g(x) + \hat{f}(x, y) - h(y)\}$, where $\hat{f}(x, y) = f(x, y) - \frac{\hat{\mu}}{2}\|y - \hat{y}\|^2$ and given arbitrary $\hat{y} \in dom\ h$. Based on Theorem 1 and careful selection of $\hat{\mu}$, Theorem 2 ensures that ZO-GDEGA can find an $\epsilon$-stationary point for the NC-C problem (1).

**Theorem 2.** *Under Assumptions 1 and 2, our ZO-GDEGA applied to the approximate NC-SC model with $\hat{\mu} = \mathcal{O}(\epsilon^2/(\ell D_h^2))$, can guarantee to generate an $\epsilon$-stationary point $x_\epsilon$ for the NC-C problem (1), i.e., $\mathbb{E}\|\nabla\psi_{1/2\ell}(x_\epsilon)\| \leq \epsilon$, with overall complexity $\mathcal{O}((d_x + d_y)\epsilon^{-6})$ and $\mathcal{O}(d_x\epsilon^{-8} + d_y\epsilon^{-10})$ for the deterministic and stochastic settings, respectively.*

Based on stronger Definition 2, Theorem 2 ensures that our ZO-GDEGA can achieve lower complexity for NC-C problems without relying on any extra assumptions, such as continuity and decreasing sequence assumptions, whereas [50, 51] depend on at least one of them. Besides, Theorem 2 inherits the advantages of our analysis for Theorem 1, i.e., enhancing robustness and reducing complexity.

### 5.3.2 Better Bounds with Continuity-dependent.

We also analyze our ZO-GDEGA algorithm for solving the NC-C problem (1) under the Lipschitz continuity assumption, which is common in NC-C optimization such as [31, 35, 43, 4].

**Assumption 5** (Continuity). *$f(x, y)$ is $G$-Lipschitz continuous in $x$, i.e., for $\forall y \in dom\ h, \forall x, x' \in \mathbb{R}^{d_x}$ satisfies that $\|f(x, y) - f(x', y)\| \leq G\|x - x'\|$.*

In the case of **(ii)** in Table 1, Assumption 5 follows immediately by the smoothness and choosing $G = \ell(D_{\mathcal{X}} + D_{\mathcal{Y}}) + \|\nabla_x f(x_\epsilon, y_\epsilon)\|$, where $\max_{x,x' \in \mathcal{X}} \|x - x'\| \leq D_{\mathcal{X}}$ and $\max_{y,y' \in \mathcal{Y}} \|y - y'\| \leq D_{\mathcal{Y}}$. Thus, ZO-GDEGA still work without Assumption 5 in this case. For the general problem (1), we use $G$ for generalization and give a proof sketch. Detailed proofs are provided in the Appendix.

***Proof sketch.*** We first analyze the recursive relationship between the Moreau envelopes $\psi_{1/2\ell}(x_t)$ and $\psi_{1/2\ell}(x_{t+1})$ based on the ZO gradient descent step. Then, we estimate the tricky error term $\Delta_t = \Phi(x_t) - \Gamma(x_t, z_t)$ to measure the upper bound of the ZO EG ascent steps. Based on these results, we obtain the overall complexity for solving the NC-C problem (1). In deterministic and stochastic settings, we provide tighter results than Theorem 2, respectively.

**Deterministic Setting.** Lemma B.13 provides the recursive relationship between $\psi_{1/2\ell}(x_{t+1})$ and $\psi_{1/2\ell}(x_t)$. Our algorithms use more concise ZO estimation gradient descent step instead of EG structure to update $x$. Thus, the tricky term $\|\nabla_x f(x_t, y_t) - \nabla_x f(x_{t-1}, y_{t-1})\|^2$ can be removed in our analysis compared

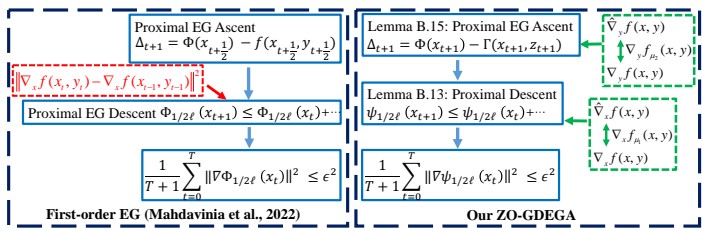

Figure 1: Comparison of two proof sketches.

with the standard EG method in [35] as shown in Fig. 1. Lemma B.15 proves that $\sum_{t=0}^T \mathbb{E}[\Delta_t]$ can be well controlled by carefully choosing $\eta_y$. ***More importantly, our analysis makes the coefficients of $\mu_1^2$ and $\mu_2^2$ independent of $\epsilon$, which eases their dependence on $\epsilon$ compared with [51], thereby enhancing the robustness.*** Now, we begin to provide tighter results.

**Theorem 3.** *Under Assumptions 1, 2 and 5, if $q_1 = d_x$, $q_2 = d_y$, $\eta_y = 1/(\sqrt{3}\ell)$, $\eta_x = \mathcal{O}(\epsilon^4)$, $\mu_1 = \mathcal{O}(\frac{\epsilon}{\sqrt{d_x}})$ and $\mu_2 = \mathcal{O}(\frac{\epsilon}{\sqrt{d_y}})$, our deterministic ZO-GDEGA can find an $\epsilon$-stationary point of $\psi$, i.e., $\frac{1}{T+1}\sum_{t=0}^T \mathbb{E}\|\nabla\psi_{1/2\ell}(x_t)\| \leq \epsilon$, for the NC-C setting with the overall complexity $\mathcal{O}((d_x + d_y)\epsilon^{-6})$.*

Theorem 3 shows that our ZO-GDEGA achieves the same overall complexity as Theorem 2 and further weakens the constraints on smoothing parameters under the continuity assumption. Note that ZO-GDEGA is the first ZO algorithm with convergence guarantees to find an $\epsilon$-stationary point of $\psi$. Meanwhile, our analysis can be easily extended to the stochastic setting.

**Stochastic Setting.** Due to the page limit, we directly provide results for stochastic ZO-GDEGA.

**Theorem 4.** *Under assumptions 1, 2, 3 and 5, if we choose the suitable parameters such as $\mu_1 = \mathcal{O}(\epsilon)$ and $\mu_2 = \mathcal{O}(\epsilon)$, our stochastic ZO-GDEGA algorithm can find an $\epsilon$-stationary point in terms of Definition 2 with lower overall complexity $\mathcal{O}(d_x\epsilon^{-6} + d_y\epsilon^{-8})$ than Theorem 2.*

Theorem 4 shows that the overall complexity can be further reduced, and we can still choose $\mu_1$ and $\mu_2$ to be $\mathcal{O}(\epsilon)$ to enhance the robustness. Note that we provide theoretical guarantees for the stochastic black-box NC-C setting for the first time. In summary, our ZO-GDEGA shows excellent theoretical properties for solving NC-C problems and competitiveness for solving NC-SC problems.

# 6 Experiments

In this section, we conduct black-box data poisoning attack and AUC maximization experiments. Due to the page limit, more experimental details and results are provided in the Appendix. Our codes are available: https://github.com/Weixin-An/ZO-GDEGA.

## 6.1 Data Poisoning Attack

Data poisoning attack is one of the most common black-box attack methods [29, 42]. The purpose of the attacker is: when the model parameter $w$ is well-trained, adding a perturbation $\delta$ on the training data that makes the loss functions as large as possible, and then such poisoned training data

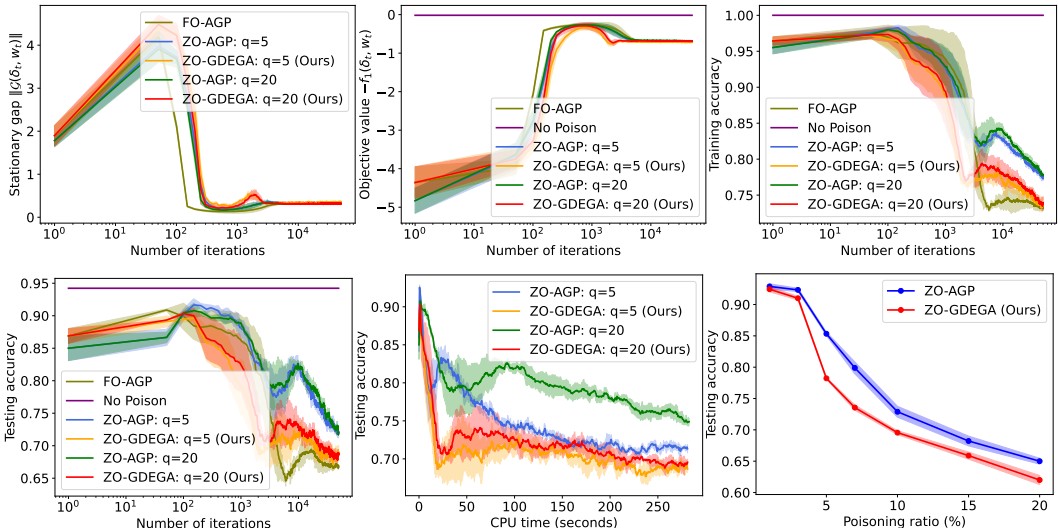

Figure 2: Comparison of the results for the logistic regression model attacked by poisoned data generated by ZO-AGP and ZO-GDEGA on the `synthetic` dataset.

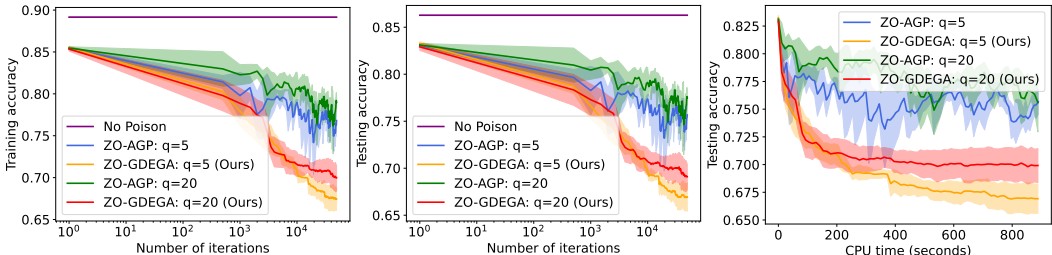

Figure 3: Comparison of the attack results for data poisoning attack on the large-scale `epsilon_test` dataset.

$(s_i + \delta, t_i)$ will reduce the model accuracy. Therefore, it is usually required for attackers to optimize the following objective function,

$$\max_{\|\delta\|_\infty \le r_x} \min_w f_1(\delta, w) := F_{tr}(\delta, w; \mathcal{D}_{tr}), \tag{5}$$

where $\mathcal{D}_{tr} \triangleq \mathcal{D}_{tr,p} \cup \mathcal{D}_{tr,c}$ denotes the training dataset including $n$ i.i.d samples, $\mathcal{D}_{tr,p}$ and $\mathcal{D}_{tr,c}$ denote the poisoned and clean subsets of $\mathcal{D}_{tr}$, respectively. We choose the cross-entropy loss to optimize $\delta$ and $w$. Note that Problem (5) can be rewritten as the NC-C (1) by setting $g(\cdot) = \mathcal{I}_{\|\delta\|_\infty \le r_x}(\cdot)$, $h(\cdot) \equiv 0$, and $f = -f_1$. Therefore, it can be solved by our ZO-GDEGA algorithm.

**Datasets.** We validate ZO-GDEGA on a `synthetic` dataset and the `epsilon_test` dataset.

● `Synthetic` datasets: We generate a dataset $\mathcal{D} = \{s_i, t_i\}_{i=1}^{2,000}$ with feature vector $s_i \in \mathbb{R}^{100}$ from Gaussian distribution $\mathcal{N}(0, I)$ and split it into 70% training and 30% test samples. The label $t_i = 1$ if $1/(1 + e^{-(s_i^\top w^* + n_i)}) > 0.5$, otherwise $t_i = 0$, where $n_i \in \mathcal{N}(0, 10^{-3})$ is random noise, and we set $w^* = \mathbf{1}$ as the ground truth.

● The `epsilon_test` dataset[3]: It contains 100,000 samples of 2,000 dimensions, and we also split it into 70% training samples and 30% testing samples.

**Experimental Settings & Baselines.** We set $r_x = 2$, $\mu_1 = \mu_2 = 2 \times 10^{-5}$ and poisoning ratio $|\mathcal{D}_{tr,p}|/|\mathcal{D}_{tr}| = 0.1$, and choose mini-batch size $b_1 = b_2 = 100$ and $b_1 = b_2 = 10$ for the `synthetic` and `epsilon_test` datasets, respectively. The baseline for solving Problem (5) is ZO-AGP. Note that here we replace the ZO estimators in deterministic ZO-AGP with corresponding stochastic versions and choose the same hyperparameter settings for a fair comparison.

**Results.** We use the poisoned data generated by the baselines and our ZO-GDEGA to attack the training procedure of the logistic regression model, and the experimental results on the `synthetic`

---

[3]https://www.csie.ntu.edu.tw/~cjlin/libsvmtools/datasets/

and `epsilon_test` datasets are shown in Figs. 2 and 3, respectively. The experiment in each case was carried out in 10 independent trials with random initialization and the shaded area around the line indicates the standard deviation. From Fig. 2, we find that our ZO-GDEGA can converge faster than baselines in terms of stationary gap and objective value $-f_1$. From the training accuracy subfigure, both our ZO-GDEGA and the baselines (black-box attack) converge, and achieve near-optimal solution compared to the FO-AGP algorithm (white-box attack) [52]. As for testing accuracy, it can be seen that compared with ZO-AGP, the poisoned data generated by our ZO-GDEGA are always more effective in reducing the accuracy of the logistic regression model in terms of both iterations and CPU time, which also confirms that our ZO-GDEGA achieves lower complexity than ZO-AGP. Besides, under different poisoning ratios, ZO-GDEGA can reduce the model accuracy by 2.1%-7.6% than ZO-AGP in terms of attack performance. From Fig. 3, we observe that ZO-GDEGA algorithm also performs about 8.0% lower than ZO-AGP on the `epsilon_test` dataset, which further verifies the superiority of our algorithm in the large-scale real-world datasets.

## 6.2 AUC Maximization

AUC is defined as the area under the ROC curve, which is a performance indicator to measure the pros and cons of a machine learning model. However, it is generally difficult to optimize the AUC value directly on the training task, but instead optimize the following minimax problem:

$$\min_{\|\theta\|, \|a\|, \|b\| \le r_x, \, v \le r_y} \max \mathbb{E}_{\mathbf{s} \sim \mathbb{P}}[f(\theta, a, b, v; \mathbf{s})], \tag{6}$$

where $r_x$ and $r_y$ are the radius of the projection balls, $\mathbf{s} = (s, t)$ is drawn independently from the distribution $\mathbb{P}$, and $f(\theta, a, b, v; \mathbf{s})$ is defined as in [53]. When we choose the multilayer perception (MLP) model, Problem (6) becomes a NC-SC problem. The non-differentiable point of the nonlinear function such as ReLU causes some non-differentiable modules of the classifier, and the ZO algorithms are one of the techniques to solve this issue [15]. Note that our ZO-GDEGA is the first ZO algorithm to attempt to solve the AUC maximization problem.

**Experimental Settings & Baselines.** We conduct experiments on the `MNIST`, `Fashion-MNIST` and `ijcnn1` datasets. Following [54], we use their training and testing sets but convert the classes of the data into two classes by randomly splitting them into two groups. We implement ZO-SGDA [47], ZO-Min-Max [33], Acc-ZOMDA [19] and our ZO-GDEGA to solve Problem (6) with Leaky ReLU. We choose a two-layer MLP as the classification model, and set mini-batch $b_1 = b_2 = 256$, $q_1 = q_2 = 10$, $r_x = r_y = 2$ and step sizes $\eta_x = \eta_y = 0.1$ to train all the methods for 200 epochs.

Table 2: The average AUC performance with different $\mu_1$ and $\mu_2$ on the `MNIST` and `Fashion-MNIST` datasets, where $\mu_1 = \mu_2$. Results on dataset `ijcnn1` are provided in the Appendix.

| Datasets $\mu_1(\mu_2)$ | MNIST | | | Fashion-MNIST | | |
|---|---|---|---|---|---|---|
| | 0.001 | 0.01 | 0.05 | 0.001 | 0.01 | 0.05 |
| ZO-SGDA | 91.67 | 91.81 | 88.12 | 91.62 | 90.19 | 87.27 |
| ZO-Min-Max | 92.25 | 92.01 | 88.56 | 90.80 | 91.58 | 83.23 |
| Acc-ZOMDA | 92.45 | 92.58 | 89.35 | 92.97 | 91.65 | 87.75 |
| ZO-GDEGA | 91.60 | 92.60 | 89.70 | 91.97 | 92.23 | 88.66 |

**Results.** We show the average AUC performance from 10 independent trials with random initialization in Table 2. It shows that at small $\mu_1$ and $\mu_2$, our ZO-GDEGA algorithms perform competitively compared with the baselines. At larger $\mu_1$ and $\mu_2$, ZO-GDEGA can improve AUC performance by 0.4-5.4 units than baselines, which indicates that our algorithms are more robust to smoothing parameters and robustness may be more important than overall complexity in some cases when using ZO oracle to approximate gradients.

## 7 Conclusions and Future Work

In this paper, we proposed a unified ZO-GDEGA algorithm for solving black-box nonconvex minimax problems, which reduces the overall complexity and improves robustness. The experimental results match our theoretical analysis, which is reflected by the fact that ZO-GDEGA can obtain more effective attacks and improve AUC performance robustly. In summary, ZO-GDEGA achieves better performance than related algorithms and promotes the development of ZO optimization theory. In the future, we will extend our algorithm to non-smooth and federated learning settings as in [45, 44].

## 8 Acknowledgments

We want to thank the anonymous reviewers for their valuable suggestions and comments. This work was supported by the National Key Research and Development Program of China (No. 2023YFF0906204), National Natural Science Foundation of China (No. 62276182), and Peng Cheng Lab Program (No. PCL2023A08).

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
