# OpenReview forum: "Robust and Faster Zeroth-Order Minimax Optimization: Complexity and Applications"
_NeurIPS.cc/2024/Conference — NeurIPS 2024 poster_

### Official Review · Reviewer_A7AZ · 2024-07-01

**Soundness:** 3
**Presentation:** 3
**Contribution:** 3
**Rating:** 6
**Confidence:** 4

**Summary:**

In this paper, a unified single-loop zeroth-order gradient descent extragradient ascent (ZO-GDEGA) algorithm to solve the nonconvex-concave minimax problem faster and more robustly. The theoretical analysis is provided to guarantee an overall complexity of $O(\epsilon^{-6})$. The experimental results on the data poisoning attack task and the AUC maximization task are shown to validate the practical performance of the proposed method.

**Strengths:**

1. In this paper, the authors propose a zeroth-order algorithm that achieves lower complexity under the nonconvex-concave condition.
2. The first convergence analysis of stochastic zeroth-order algorithm under the nonconvex-concave condition is provided.
3. For nonconvex strongly-concave problems, the complexity with respect to the condition number is improved.

**Weaknesses:**

1. In section 2 related work, the complexity of first-order minimax algorithms is not discussed. To my understanding, the error of the zeroth-order gradient can be bounded by parameters $\mu_1$ and $\mu_2$ that are set as small as $O(\epsilon)$. Therefore, the analysis should not change a lot based on the analysis of the first-order counterpart, which probably undermines the novelty and contribution.
2. In ref [18] (Huang et al 2022), variance reduction is used to improve the complexity with respect to $\epsilon$. The contribution of this paper will be further stronger if this part is also considered.
3. The figures in the experiments section is too small and some curves are covered by the legends. I understand the space is limited but maybe some sections could be rearranged to the Appendix.

**Questions:**

1. Please see weakness [1]. Can you provide more details to explain how the zeroth-order gradient estimator makes the original analysis of the first-order method more challenging?
2. Is it possible to further reduce the complexity with respect to $\epsilon$ for the NC-C problem by integrating variance reduction with the proposed ZO-GDEGA algorithm?

**Limitations:**

I did not see any negative societal impact.

---

> ### Author Rebuttal · Authors · 2024-08-07
>
> Thanks for your insightful thoughts and comments! Below we will clarify the two points in the review.
>
> ### **Response 1.**
>
> **Related works for first-order methods.**
>
> Existing works mainly focus on first-order (FO) methods for solving NC minimax problems. For the NC-SC setting, GDA [1] and AGDA [2] both achieve $\mathcal{O}(\kappa^2\epsilon^{-2})$ ($\mathcal{O}(\kappa^3\epsilon^{-4})$) gradient complexity in deterministic (sochastic) setting to find an $\epsilon$-stationary point. Lin et.al [3] proposed a multi-loop Minimax-PPA algorithm, which further improves the complexity to $\mathcal{O}(\sqrt{\kappa}\epsilon^{-2})$. For the NC-C setting, GDA [1] and AGDA [2] both achieve a gradient complexity of $\mathcal{O}(\epsilon^{-6})$ and $\mathcal{O}(\epsilon^{-8})$ for the deterministic and stochastic settings. SAPD+ [4] has reduced the gradient complexity to $\mathcal{O}(\epsilon^{-6})$ for the stochastic setting. The above analysis is all measured with the max function $\Phi$. Xu et.al [5] considered the $\epsilon$-stationary point in terms of $f$ and proposed AGP  to achieve a gradient complexity of $\mathcal{O}(\epsilon^{-4})$ on this weaker measure, but did not consider the stochastic setting. FO algorithms inspired the design of ZO algorithms. However, existing ZO methods such as [5][6] mainly focus on trivial extensions of existing FO algorithms, while we focus on designing ZO minimax algorithms and directly improving the robustness and convergence rate, which is a non-trivial extension.
>
> **In our main paper, we stated the challenges. To address your concerns about our contributions, we clarify them and improve discussion of challenges as follows.**
>
> We propose a faster, more robust, and theoretically guaranteed algorithm framework that can solve wider applications.
>
> **Stronger robustness.** We find that the smoothing parameter bounds have a key impact on the ZO algorithms for black-box minimax problems. Previous research did not focus on this key issue. We designed a novel ZO-GDEGA algorithm to weaken restrictions on the smoothing parameters (see Table 1), which enhances the robustness. Our ZO-GDEGA has also been extended to novel FO methods, which are shown in Algorithms 3 and 4 in the Appendix. Thus, our algorithm is a **non-trivial** extension of the existing FO algorithms.
>
> **Lower complexity.** We develop a new and concise analysis framework to prove that our ZO-GDEGA can achieve lower complexity and it is the first ZO algorithm with theoretical guarantees for solving stochastic NC-C problems.
>
> **Analysis of Challenges.** For minimax problems, the errors $||\hat\nabla_x f(x,y) - \nabla_x f(x,y)||$ and $||\hat\nabla_y f(x,y) - \nabla_y f(x,y)||$ are coupled with each other, which leads to difficult parameter selection. Taking the NC-C setting as an example, the recursion $||\hat x_{t+1} - x_t||$ requires $||\hat\nabla_y f(x,y) - \nabla_y f(x,y)||$ and $\Delta_t$ requires $||\hat\nabla_x f(x,y) - \nabla_x f(x,y)||$. We handle them by constructing reasonable recursions and step sizes. Our Proposition 6 plays a key role and clarifies that the EG structure can result in the coefficient $\mathcal{O}(1)$ for $\mu_2$ and $\mathcal{O}(\eta_x^2\eta_y)$ for $\mu_1$, which allows to choose reasonable step sizes and larger smoothing parameters than existing methods, thus achieving more robust performance.
>
> Even our analysis for the FO counterpart still faces difficulties due to the EG and the proximal operator. In the NC-SC setting, we propose to establish the upper bound on $\sum_{t=0}^T\delta_t$ in terms of $\sum_{i=0}^T||x_i - x_{i+1}||^2$ to construct the recursive relation of $y$, thereby achieving enhanced robustness. In the NC-C setting, we creatively proposed Proposition 6 to handle the proximal operator and EG structure, thereby resulting in the reasonable recursion w.r.t. $y$.
>
> **Wider applications.** We expand the scope of existing black-box minimax problems by considering more general regularizers rather than being limited to $g=\mathcal{I_X},f=\mathcal{I_Y}$, which can be instantiated as data poisoning attack, AUC maximization, and robust neural network training (see C.3 in the Appendix) problems.
>
> **Better experimental results.** Our algorithms can find much more effective attack perturbation $\delta$ and perform more robustly.
>
> ### **Response 2.  Integrating variance reduction with the proposed ZO-GDEGA can further reduce the complexity.**
>
> Taking the NC-SC as an example, we provide an idea. We can use the ZO recursive momentum variance-reduced stochastic gradients as follows:
> $$
> u_t = \alpha_t \hat\nabla_x f(x_t, y_t; \xi)  + (1-\alpha_t)(\hat\nabla_x f(x_t, y_t;\xi) - \hat\nabla_x f(x_{t-1}, y_{t-1}; \xi) + u_{t-1}),
> $$ to update $x$, as well as $y$, where $\alpha_t \in (0, 1]$. When $\alpha_t = 1$, $u_t$ will degenerate a vanilla ZO stochastic gradients. In theoretical analysis, since STORM's analysis [7] can effectively reduce the complexity with respect to $\epsilon$, we can refer to their analysis and design Lyapunov function $\phi = f(x_t, y_t) + \mathcal{O}(\frac{1}{L^2\eta_x})||u_t - \nabla_x f(x_t,y_t)||^2 + \mathcal{O}(\frac{1}{L^2\eta_y})||v_t - \nabla_y f(x_t,y_t)||^2$ to analyze the complexity, and thus the complexity can be reduced. As for the NC-C setting, we will also think about how to design Lyapunov functions to analyze complexity.
>
> We will rearrange tables and figures in our revised version.
>
> [1] On gradient descent ascent for nonconvex-concave minimax problems
>
> [2] Alternating proximal-gradient steps for (stochastic) nonconvex-concave minimax problems[J]
>
> [3] Near-optimal algorithms for minimax optimization
>
> [4] Sapd+: An accelerated stochastic method for nonconvex-concave minimax problems.
>
> [5] DERIVATIVE-FREE ALTERNATING PROJECTION ALGORITHMS FOR GENERAL NONCONVEX-CONCAVE MINIMAX PROBLEMS
>
> [6] Enhanced first and zeroth order variance reduced algorithms for min-max optimization[J]. 2020.
>
> [7] Momentum-based variance reduction in non-convex sgd

---

> > ### Comment · Reviewer_A7AZ · 2024-08-13
> > **Official Comment by Reviewer A7AZ**
> >
> > Thank you very much for the reply. After reading the rebuttal, I think my concerns are addressed and I have raised my rating to 6.

---

> > > ### Author Response · Authors · 2024-08-13
> > >
> > > We appreciate your review and valuable questions that improved our work.

---

### Official Review · Reviewer_4438 · 2024-07-08

**Soundness:** 3
**Presentation:** 3
**Contribution:** 3
**Rating:** 6
**Confidence:** 2

**Summary:**

They design a new unified ZO gradient descent extragradient ascent (ZO-GDEGA) algorithm, which reduces the overall complexity to find an ε-stationary point of the function ψ for nonconvex-concave (NC-C) problems. ZO-GDEGA is the first ZO algorithm with complexity guarantees to solve stochastic NC-C problems.

**Strengths:**

The theoretical section is very thorough and comprehensive.

**Weaknesses:**

(1)The paper lacks detailed link of code implementation.
(2)While the paper effectively addresses the NC-C problem from a theoretical perspective, it should also provide more experimental details and demonstrate that the NC-C problem exists within the experimental models.

**Questions:**

See in the Weakness part

---

> ### Author Rebuttal · Authors · 2024-08-07
>
> ### **Response 1. The detailed link of code.**
>
> We provide the code of the data poisoning attack experiment on the epsilon\_test dataset. We have sent this code sample as an anonymized link to the AC. We will make our code public.
>
> ### **Response 2. The proof of the NC-C problem (5).**
>
> Problem (5) is a NC-C problem which can be proved as follows.
>
> For Problem (5) (please see the C.1 in the Appendix for detail), we let $\mathcal{L}(\delta,w;s_i) = \frac{1}{1 + e^{-(s_i + \delta)^\top w}}$ and $\mathcal{G}(w):= t_i\log(\mathcal{L}(\delta,w;s_i)) + (1-t_i)(\log (1 - \mathcal{L}(\delta,w;s_i)))$. We prove that the function $\mathcal{G}(w)$ is general convex as follows.
> $$\frac{\partial \mathcal{G}}{\partial w} = t_i\frac{1}{\mathcal{L}}\frac{\partial \mathcal{L}}{\partial w} - (1-t_i) \frac{1}{1-\mathcal{L}}\frac{\partial \mathcal{L}}{\partial w} = t_i(1+e^{-(s_i + \delta)^\top w})\frac{(s_i + \delta)e^{-(s_i + \delta)^\top w}}{(1+e^{-(s_i + \delta)^\top w})^2} - (1-t_i)\frac{1 + e^{-(s_i + \delta)^\top w}}{e^{-(s_i + \delta)^\top w}}\frac{(s_i + \delta)e^{-(s_i + \delta)^\top w}}{(1+e^{-(s_i + \delta)^\top w})^2} $$ $$= t_i\frac{(s_i + \delta)e^{-(s_i + \delta)^\top w}}{1+e^{-(s_i + \delta)^\top w}} - (1-t_i) \frac{s_i + \delta}{1+e^{-(s_i + \delta)^\top w}} = t_i((s_i + \delta) - \frac{s_i + \delta}{1+e^{-(s_i + \delta)^\top w}}) - (1-t_i) \frac{s_i + \delta}{1+e^{-(s_i + \delta)^\top w}}$$
> $$ = t_i(s_i + \delta) - \frac{s_i + \delta}{1 + e^{-(s_i + \delta)^\top w}}.$$ Furthermore, $\frac{\partial^2 \mathcal{G}}{\partial^2 w} = \frac{(s_i + \delta)^2e^{-(s_i + \delta)^\top w}}{(1+e^{-(s_i + \delta)^(s_i + \delta)^\top w})^2} \geq 0$. This Hessian matrix is positively semi-definite. Thus, the function $\mathcal{G}(w)$ is general convex and $f_1(\delta, w)$ is general convex w.r.t. $w$. Thus, Problem (5) can be written as the NC-C formula (1) by setting $g(\cdot)=I_{\Vert\delta\Vert_{\infty}\leq r_x}$,  $h(\cdot) \equiv 0$, and  $f = -f_1$.
>
> For more experimental details, please refer to the Appendix, where we describe the selection of hyperparameters and more experimental results and analysis.

---

> > ### Author Response · Authors · 2024-08-14
> >
> > We appreciate your review and valuable questions that improved our work.

---

### Official Review · Reviewer_hV36 · 2024-07-12

**Soundness:** 3
**Presentation:** 3
**Contribution:** 3
**Rating:** 6
**Confidence:** 2

**Summary:**

The paper studies zeroth-order methods for nonconvex-(strongly)-concave minimax optimization. The achieved rates improve previous results and tolerate much larger choice of the smoothing parameters. The proposed methods also perform well for some empirical tasks.

**Strengths:**

Minimax optimization is an important problem that has many applications in machine learning and related areas. In many settings, gradients are hard to estimate or impossible to obtain, which motivates the study of zeroth-order methods. Moreover, nonconvex-concave minimax is itself a class of nonconvex nonsmooth optimization, which is considered as challenge problems in the related literature. The paper proposes new algorithms with improved convergence results compared with previous work for this challenge class of the problem.

**Weaknesses:**

1. I think a discussion on lower bounds can improve the understanding and position of this work. For example, lower bounds on zeroth-order methods justify the dependence on the dimension is inevitable without additional assumptions [1]. This, together with lower bounds for minimax optimization, e.g., [2], provide lower bounds for the considered problem class, which suggest the foundamental limits of this problem and whether the complexity can be further improved.

2. A discussion of the best known upper bounds for first-order methods could also help. As far as I know, the best rate for first-order nonconvex-concave minimax optimization is also $\epsilon^{-6}$ [3]. The authors can do some additional literature review and check whether my statement is correct. As the complexity of zeroth-order methods is usually d times that of first-order methods, this suggests the results in this paper match the state-of-the-arts. For nonconvex-strongly-concave minimax optimization, [3] achieves a rate of $\kappa\epsilon^{-4}$, which suggests that the upper bounds in this paper can possibly be improved.

3. Have the authors considered to use two-point estimators to construct zeroth-order gradient estiimators? In some cases, two-point estimators give better rates [4]. The paper mentions that for the NC-SC case there is no need to use $z_t$ to update $x_{t+1}$ but $y_t$ instead. However, there is $z_t$ involved in the update of $y_t$. Are there any typos in the statement of the algorithm? Also, I suggest to add dimension in the complexity stated in the abstract. Otherwise, this could be misleading.

I did not have time to carefully check every step in the proof. If the results are correct, I think they make enough contributions to the related literature. Therefore, I will keep a low confidence score.

References

[1] Optimal rates for zero-order convex optimization: The power of two function evaluations. IEEE Transactions on Information Theory, 2015.

[2] The complexity of nonconvex-strongly-concave minimax optimization. UAI, 2021.

[3] SAPD+: An Accelerated Stochastic Method for Nonconvex-Concave Minimax Problems. NeurIPS, 2022.

[4] An optimal algorithm for bandit and zero-order convex optimization with two-point feedback. JMLR, 2017.

**Questions:**

See weaknesses.

**Limitations:**

See weaknesses.

---

> ### Author Rebuttal · Authors · 2024-08-07
>
> We would to thank the reviewer for the insightful comments! Below we will clarify the four points in the review.
>
> ### **Response 1. Lower Bounds.**
>
> We will add the following discussion about the lower bound to our revised version.
>
> For minimization problems, the lower bounds on ZO methods justify the dependence on the dimension is inevitable without additional assumptions [1]. For minimax problems, some researchers focus on the lower bound of first-order methods. For example, the lower bounds are $\Omega(\epsilon^{-1})$ [2] and $\widetilde\Omega(\kappa)$ [3] for convex-concave and strongly-convex-strongly-concave settings, respectively.  For NC-SC minimax problems, there is a lower bound $\mathcal{O}(\sqrt{\kappa}\epsilon^{-2})$ of first-order methods [4][5]. Corresponding to minimization optimization [1], there may be a lower bound $\mathcal{O}((d_x + d_y)\sqrt{\kappa}\epsilon^{-2})$ for ZO minimax optimization. Unfortunately, to the best of our knowledge, there is no work proving a lower bound for ZO algorithms solving NC-SC minimax problems. On the contrary, our work provides the upper bound for ZO algorithms solving NC minimax problems. Thus, more research is needed.
>
> ### **Response 2. Upper Bounds.**
>
> For the NC-C setting, GDA [6], AGDA [7] and GDmax [8, 9] are analyzed to guarantee convergence. To the best of our knowledge,  the best complexity for first-order deterministic NC-C optimization is $\mathcal{O}(\epsilon^{-6})$. The works [10, 11] have further extended to the stochastic setting and achieved the complexity matching that of the deterministic setting. For example, SAPD+ [10] achieved the gradient complexity of $\mathcal{O}(L^3\epsilon^{-6})$. As the complexity of ZO methods is usually $d$ times that of first-order methods [1], in this sense, it suggests our results $\mathcal{O}((d_x + d_y)\epsilon^{-6})$ match the state-of-the-arts in the deterministic setting.
>
> For the stochastic NC-SC setting, SGDA, SAGDA and SODGA all achieved the gradient complexity of $\mathcal{O}(\kappa^3\epsilon^{-4})$. The works [12][13] reduced the complexity to $\mathcal{O}(\kappa^3\epsilon^{-3})$. Recently, SAPD+ has achievesd a complexity of $\mathcal{O}(\kappa\epsilon^{-4})$, which suggests that the upper bounds in this paper can possibly be improved. Unfortunately, there is no similar conclusion as in [1] for NC minimax problems so far. Moreover, existing ZO methods such as [13][14] mainly focus on trivial extensions of existing first-order algorithms, while this paper focuses on designing ZO minimax algorithms and directly improving the robustness and convergence rate, which is a non-trivial extension.
>
> We will add the discussion of upper bounds for first-order methods to our revised version.
>
> ### **Response 3. Other Questions.**
>
> To address your concern, we try to analyze our algorithm under two-point estimation, and we find that our algorithm still maintains the advantages of two-point estimation, that is, the second moment of gradient estimate is essentially linear in the dimension $d_x + d_y$. In our revised version, we will clarify this.
>
> Updating $x_{t+1}$ does not require the intermediate point $\{z_t\}$, but the sequence $\{z_t\}$ is needed as an intermediate point to update $y$. Thus, this is not a typo.
>
> Besides, we will add dimension to the complexity stated in the abstract of the revised version.
>
> ### **Response 4. Our proof is correct.**
>
> After our careful inspection, we believe that all our proofs are correct.
>
> [1] Optimal rates for zero-order convex optimization: The power of two function evaluations[J]. IEEE Transactions on Information Theory, 2015, 61(5): 2788-2806.
>
> [2] Lower complexity bounds of first-order methods for convex-concave bilinear saddle-point problems[J]. Mathematical Programming, 2021, 185(1): 1-35.
>
> [3] On lower iteration complexity bounds for the saddle point problems[J]. arXiv preprint.—2018.—https://arxiv. org/pdf, 1812.
>
> [4] The complexity of nonconvex-strongly-concave minimax optimization[C]//Uncertainty in Artificial Intelligence. PMLR, 2021: 482-492.
>
> [5] Complexity lower bounds for nonconvex-strongly-concave min-max optimization[J]. Advances in Neural Information Processing Systems, 2021, 34: 1792-1804.
>
> [6] On gradient descent ascent for nonconvex-concave minimax problems[C]//International Conference on Machine Learning. PMLR, 2020: 6083-6093.
>
> [7] Alternating proximal-gradient steps for (stochastic) nonconvex-concave minimax problems[J]. SIAM Journal on Optimization, 2023, 33(3): 1884-1913.
>
> [8] Minimax optimization: stable limit points of gradient descent ascent are locally optimal (2019). arXiv preprint arXiv:1902.00618.
>
> [9] What is local optimality in nonconvex nonconcave minimax optimization? In International conference on machine learning, pages  4880–4889. PMLR, 2020.
>
> [10] Sapd+: An accelerated stochastic method for nonconvex-concave minimax problems. Advances in Neural Information Processing  Systems, 35:21668–21681, 2022.
>
> [11] Adagda: Faster adaptive gradient descent ascent methods for minimax optimization. In International Conference on Artificial Intelligence and Statistics, pages 2365–2389. PMLR, 2023.
>
> [12] Stochastic recursive gradient descent ascent for stochastic nonconvex-strongly-concave minimax problems[J]. Advances in Neural Information Processing Systems, 2020, 33: 20566-20577.
>
> [13] Enhanced first and zeroth order variance reduced algorithms for min-max optimization[J]. 2020.
>
> [14] Zeroth-order alternating randomized gradient projection algorithms for general nonconvex-concave minimax problems[J]. arXiv preprint arXiv:2108.00473, 2021.

---

> > ### Comment · Reviewer_hV36 · 2024-08-12
> >
> > Thanks for your detailed response! I will increase my score to 6 because of these detailed discussions and the efforts you took during the rebuttal phase. I have no further comments.

---

> > > ### Author Response · Authors · 2024-08-13
> > >
> > > We appreciate your review and valuable questions that improved our work.

---

### Official Review · Reviewer_Rz6B · 2024-07-13

**Soundness:** 4
**Presentation:** 3
**Contribution:** 2
**Rating:** 6
**Confidence:** 5

**Summary:**

This paper proposes zeroth-order method called ZO-GDEGA to find a near-stationary point for nonconvex-concave minimax optimization, with complexity guarantee. The proposed method is also extended to stochastic setting, being the first work on ZO method on stochastic NC-C problem. The method has weaker requirement on ZO gradient estimate, thus also has better dependency on condition number in the special case of NC-SC. Numerical results on data poisoning attack and AUC maximization show the proposed method is comparable and usually slightly better than compared baselines.

**Strengths:**

1. This work designs a unified single-loop ZO method for NC-C minimax, with better complexity and more robust allowance on ZO gradient estimate. The proposed idea of continuous-time dynamics to assist with updates of dual variable and its related analysis is novel. Overall complexity is proposed, with solid and rigorous proof, under weak assumptions such as not requiring lipschitz continuity, more tolerant ZO gradient, which also results in good complexity in the NC-SC special case.

2. The proposed method can be extended to stochastic setting, with first-ever complexity in this case.

**Weaknesses:**

1. Complexity (deterministic NC-C): The work claims the $O(d\epsilon^{-6})$ complexity of the proposed method is a 'reduced' complexity, however to the best of my knowledge, this is not the best-known complexity of ZO method on NC-C minimax. Even for single-loop methods, the existing method in [43] shown by table 1 has a better $O(d\epsilon^{-4})$ complexity. Intuitively, only a complexity as good as this existing $O(d\epsilon^{-4})$ is near-optimal, because first-order methods on NC-C minimax have $O(\epsilon^{-4})$ complexity. Although assumption-wise, as this work claims, [43] has the extra assumption of 'decreasing regular parameter sequence', and also requires more accurate ZO gradient on the primal variable, but in my opinion, the extra assumption above is not too restrictive, and requiring more accurate ZO gradient would only cost a constant multiple of queries thus would not affect overall complexity. Therefore, the $O(d\epsilon^{-6})$ complexity in this work is not good enough, and is in fact worse than certain existing single-loop ZO methods.

2. Complexity (stochastic NC-C): I acknowledge this is the first-ever complexity of ZO method on stochastic NC-C minimax. However, the
$O(d_x \epsilon^{-6} + d_y \epsilon^{-8})$ dependence is not near-optimal, since first-order methods on such problem just have $O( \epsilon^{-6})$ complexity. ** Update: typo fixed, from $O(d \epsilon^{-6})$ to $O(\epsilon^{-6})$.

3. Listed numerical results show the proposed method is similar and generally only slightly better than compared baselines. The difference is not significant in both experiments.

4. Overall, both theoretical complexities and numerical results does not surpass existing methods, thus the contribution may not be strong enough for NeurIPS standard.

**Questions:**

1. Please address [Weaknesses 1].

2. Please address [Weaknesses 2].

---

> ### Author Rebuttal · Authors · 2024-08-07
>
> We would like to thank the reviewer for the insightful comments! Below we will clarify the three points in the review.
>
> ### **Response 1. Complexity (deterministic NC-C)**
>
> $\bullet$ **Our ZO-GDEGA achieved the reduced complexity.** The key reason for different complexity is different complexity metrics. That is, our method can achieve the complexity of $\mathcal{O}(\epsilon^{-6})$ under a more difficult convergence criterion compared with [43]. As we discussed in the tablenote $4$ of Table 1, the work [43] achieved an $\epsilon$-stationary point of $f$. According to [27, Proposition 4.12] and our Propositions 1 and 2, the $\epsilon$-stationary point in terms of $f$ is weaker than that of $\psi$. Thus, [43] found a weaker $\epsilon$-stationary point with lower gradient complexity. For a fair comparison, we can convert their complexity in terms of $f$ into complexity in terms of $\psi$, that is, **[43] requires the overall complexity of $\mathcal{O}((d_x + d_y)\epsilon^{-8})$ to find an ${\epsilon}$-stationary point of $\psi$. In this sense, our ZO-GDEGA algorithm reduced the complexity of existing methods.**
>
> $\bullet$ **Assumptions.** In addition to satisfying the assumption `decreasing regular parameter sequence', the work [43] requires smaller smoothing parameters (i.e., more accurate ZO gradient) to achieve their gradient complexity. There are two dilemmas. Firstly, the constant multiple of queries may require a lot of experimental verification. Secondly, although satisfactory accuracy ZO gradient can be achieved through constant multiple of queries, smoothing parameters are required to the order of $\mathcal{O}(\epsilon^{2})$, which is more sensitive than that of our ZO-GDEGA, which means weaker robustness than our methods. In summary, [43] requires the assumption ``decreasing regular parameter sequence'' and has weaker robustness. In the experiment, we also found that the ZO-AGP [43] is more sensitive to smoothing parameters than our ZO-GDEGA. Please see Fig. C.8 in the Appendix for details.
>
> ### **Response 2. Complexity (stochastic NC-C)**
>
> To the best of our knowledge, the state-of-the-art complexity of the first-order algorithm is $\mathcal{O}(\epsilon^{-6})$ [1][2] for solving stochastic NC-C problems, but there is currently no lower bound for ZO algorithms solving stochastic NC-C problems. One of the contributions of this paper is that our ZO-SGDA **firstly** achieves the complexity of $\mathcal{O}(d_x\epsilon^{-6} + d_y\epsilon^{-8})$ in the stochastic NC-C setting. As the complexity of ZO methods is usually $d$ times that of first-order methods [3], there may be a gap for NC minimax problems. We will bridge this gap by recursive momentum variance reduction technology [4] in our future work.
>
> ### **Response 3. Significant improvements in theory and experiment**
>
> In the data poisoning attack experiment, our ZO-GDEGA can reduce the accuracy by about 2\%-8\% compared with the baseline, which is a significant improvement. In this experiment, **the lower the accuracy, the better the algorithm performs**. More experimental results and analysis are provided in the Appendix. In summary, our theoretical complexities and numerical results outperform existing methods.
>
> [1] Non-convex min-max optimization: Provable algorithms and applications in machine learning[J]. ArXiv, abs/1810.02060, 2018.
>
> [2] Sapd+: An accelerated stochastic method for nonconvex-concave minimax problems[J]. Advances in Neural Information Processing Systems, 2022, 35: 21668-21681.
>
> [3] Optimal rates for zero-order convex optimization: The power of two function evaluations[J]. IEEE Transactions on Information Theory, 2015, 61(5): 2788-2806.
>
> [4] Momentum-based variance reduction in non-convex sgd[J]. Advances in neural information processing systems, 2019, 32.

---

> > ### Comment · Reviewer_Rz6B · 2024-08-09
> > **Response to authors**
> >
> > I have read the authors' response. I suggest those valuable discussions could be added in appendix to clarify each contribution. While I agree with the theoretical claims in the rebuttal, however each weakness in my original comment still holds. Thus I will keep my score.

---

> ### Author Response · Authors · 2024-08-13
>
> Thanks for your comments again. We will add the above valuable discussions in our revised version. **We try again to address the weaknesses you mentioned. We further clarified our contributions, and stated our challenges**.
>
> ### **Addressed the weakness**
>
> **Deterministic NC-C.** Under the complexity metric $\psi$, the state-of-the-art gradient complexity is $\mathcal{O}(\epsilon^{-6})$ in the deterministic setting [27]. Thus, our ZO-GDEGA achieves the near-optimal gradient complexity $\mathcal{O}(d_x + d_y)\epsilon^{-6}$ in terms of metric $\psi$. As a comparison, [43] achieves the gradient complexity $\mathcal{O}(d_x + d_y)\epsilon^{-8}$ in terms of metric $\psi$. In this sense, our ZO-GDEGA is near-optimal and achieved the 'reduced' complexity.
>
> **Stochastic NC-C.** One of our main contributions is that we **first** provide the provable complexity framework in the stochastic NC-C setting. Although near-optimal complexity may not (because to the best of our knowledge, no work proves that the complexity $\mathcal{O}(d\epsilon^{-6})$ can be achieved) be reached, our analysis opens the door to analyzing the ZO NC-C setting.
>
> ### **Our contributions**
>
> We propose a faster, more robust, and theoretically guaranteed algorithm framework that can solve wider minimax applications.
>
> **Stronger robustness.** Developing a ZO minimax algorithm for stronger robustness is challenging, as existing ZO algorithms require a demanding ZO estimate of the gradients, i.e., very small (e.g., $\mathcal{O}(\epsilon^{-3})$ [18] [42], $\mathcal{O}(\epsilon^{-2})$ [43]) smoothing parameters. We find that the smoothing parameter bounds have a key impact on the ZO algorithms for black-box minimax problems, which is shown in C.1.2 in the Appendix. Previous research did not focus on this key issue. We designed a novel ZO-GDEGA algorithm to weaken restrictions on the smoothing parameters (see Table 1) and offer the first provable guarantee. Our ZO-GDEGA has also been extended to novel FO methods, which are shown in Algorithms 3 and 4 in the Appendix. Thus, our algorithm is a non-trivial extension of the existing algorithms.
>
> **Lower complexity.** We develop a new and concise analysis framework to prove that our ZO-GDEGA can achieve lower complexity under the same complexity metrics. It is the **first** ZO algorithm with theoretical guarantees for solving stochastic NC-C problems. **Algorithms without theoretical guarantees are unstable and untrustworthy**. The stochastic version of ZO-AGP [5] has no provable complexity guarantees. Specifically, in the proof of deterministic ZO-AGP, some parameters are severely coupled, and the stochastic setting will further complicate its analysis, while our proof framework can be easily extended to the stochastic setting.
>
> **Wider applications.** We expand the scope of existing black-box minimax problems by considering more general regularizers rather than being limited to $g=\mathcal{I}_x$ and $f=\mathcal{I}_y$, which can be instantiated as data poisoning attack, AUC maximization, and robust neural network training (see C.3 in the Appendix) problems.
>
> **Better experimental results.** Our algorithms can find much more effective attack perturbation $\delta$ and perform more robustly.
>
> ### **Analysis of Challenges**
>
> We propose a novel complexity analysis framework for ZO minimax optimization. For minimax problems, the errors $||\hat\nabla_x f(x,y;\zeta) - \nabla_x f(x,y)||$ and $||\hat\nabla_y f(x,y;\xi) - \nabla_y f(x,y)||$ are coupled with each other, which leads to difficult parameter selection. Taking the NC-C setting as an example, the recursion $||\hat{x}_{t+1} - x_t||$ requires the error $||\hat\nabla_y f(x,y;\xi) - \nabla_y f(x,y)||$ and $\Delta_t$ requires the error $||\hat\nabla_x f(x,y;\zeta) - \nabla_x f(x,y)||$. We handle them by constructing reasonable recursions and step sizes. Our **Proposition 6** plays a key role and clarifies that the EG structure can result in the coefficient $\mathcal{O}(1)$  for $\mu_2$ and $\mathcal{O}(\eta_x^2\eta_y)$ for $\mu_1$, which allows to choose reasonable step sizes and larger smoothing parameters than existing methods, thus achieving more robust performance.
>
> Even our analysis for the FO counterpart still faces difficulties due to the EG and the proximal operator. In the NC-SC setting, we propose to establish the upper bound on $\sum_{t=0}^T\delta_t$ in terms of $\sum_{i=0}^T||x_i - x_{i+1}||^2$ to construct the recursive relation of $y$ in **Lemma B.9**, thereby achieving enhanced robustness. In the NC-C setting, we creatively proposed **Proposition 6** to handle the proximal operator and EG structure, thereby resulting in the reasonable recursion w.r.t. $y$.
>
> [5] Derivative-free alternating projection 490 algorithms for general nonconvex-concave minimax problems. 2023.

---

> ### Comment · Reviewer_Rz6B · 2024-08-13
> **Response to authors**
>
> Thanks for the detailed responses. I agree the proposed methods improves the complexities compared to existing zeroth-order methods, and that is good. My point of weakness, which still holds, was that the proposed complexities are not tight, worse than d * first-order complexities; where existing first-order methods have $\epsilon^{-4}$ complexity for deterministic NC-C and $\epsilon^{-6}$ complexity for stochastic NC-C. I hope the authors understand this point, and potentially tighten those zeroth-order bounds in future works.
> With that being said, I carefully read the detailed analysis and additional experiments in the appendix, which made good efforts in making the theory and experiments comprehensive, and the analysis is nontrivial. Thus, I have increased my score from 5 to 6.

---

> > ### Author Response · Authors · 2024-08-13
> >
> > We appreciate your review and valuable questions that improved our work.

---

### Official Review · Reviewer_JCfr · 2024-07-13

**Soundness:** 3
**Presentation:** 3
**Contribution:** 2
**Rating:** 5
**Confidence:** 3

**Summary:**

In this paper, the authors establish a unified framework of zeroth-order optimization for nonconvex-concave minimax optimization problems in both deterministic and stochastic settings. This framework is based on the gradient descent-extragradient ascent algorithm. They claim that their algorithms require weaker assumptions on the zeroth-order estimator, while achieve competitive iteration complexity compared with the existing work. Besides, they provide some numerical experiments to verify the effectiveness of the proposed algorithms.

**Strengths:**

Main advantages can refer to the Summary. Besides, this paper has a good organization, which makes the readers reading easily. Numerous experiments including abundant baselines and datasets are provided to illustrate the effectiveness of the proposed algorithm.

**Weaknesses:**

In the stochastic setting, the authors make the bounded variance assumption on the zeroth-order estimator in Assumption 3. It seems that the assumption makes the proof very similar to first-order method. It is true that the zeroth-order methods are used when computing derivative are not possible, but does this assumption rules out the most challenging technical part in the proof?
This paper has proposed zeroth-order method that is motivated by application scenes when the objective function is not smooth enough to enable one to compute the gradient. But there is no specific examples provided that such method plays a prominent role while the first-order method is not applicable. Since there is a definite trade-off when zeroth-order method are used compared to first-order, the smooth parameter is another factor that should be considered in using zeroth-order method. So I think providing a concrete application that zeroth-order method is inevitable will help convincing the importance of the algorithm in application.

**Questions:**

Similar to aforementioned concerns on example, the experiments of poisoning data attack is a clear application that zeroth-order can be used. My question is whether it is possible to use first-order method in some cases? If this is the application that no first-order method can be used, then my previous conern on weakness can be addressed. I don't here mean that the authors should have compare zeroth-order method to first-order method in performance, since it is completely reasonable that first-order method outperforms zeroth-order and even it holds, nothing negative will impact the contribution of the paper.

**Limitations:**

This paper is mostly a theoretical work, no negative societal impact may result in.

---

> ### Author Rebuttal · Authors · 2024-08-07
>
> We would like to thank the reviewer for the insightful thoughts and comments! Below we will clarify the two points in the review.
>
> ### **About Assumption 3.**
>
> Firstly, analyzing ZO algorithms has the following two main difficulties compared to analyzing first-order methods. That is, we need to bound two error terms, the error $||\hat\nabla_x f(x,y;\zeta) - \nabla_x f_{\mu_1}(x,y)||$ (or $||\hat\nabla_y f(x,y;\xi) - \nabla_y f_{\mu_2}(x,y)||$) and $||\nabla_x f_{\mu_1}(x,y) - \nabla_x f(x,y)||$ (or $||\nabla_y f_{\mu_2}(x,y) - \nabla_y f(x,y)||$). We first bound the first error by Assumption 3. The same assumption also appears in the classic work [1]. In this paper, we mainly focus on solving the second difficulty. We handle the second error of the ZO EG estimates by constructing reasonable recursions. Taking the NC-C setting as an example, our Proposition 6 plays a key role and clarifies that the EG structure can result in the coefficient $\mathcal{O}(1)$ for $\mu_2$ and $\mathcal{O}(\eta_x^2\eta_y)$ for $\mu_1$ (please see the inequality B.130 for detail), which allows us to choose larger smoothing parameters (i.e., $\mu_1 = \mathcal{O}(\epsilon)$ and $\mu_2 = \mathcal{O}(\epsilon)$) than existing methods, thus achieving more robust performance.
>
> Secondly, due to the introduction of EG and proximal operator in our ZO-GDEGA, our first-order analysis framework still faces difficulties. That is, the EG and proximal operator of our algorithm make the analysis framework more difficult. Taking the NC-SC setting as an example, we propose to establish the upper bound on $\sum_{t=0}^T\delta_t$ in terms of $\sum_{i=0}^T||x_i - x_{i+1}||^2$ to construct a recursive relation, thereby achieving competitive complexity and enhanced robustness.
>
> In summary, there are still many challenges in our ZO theoretical analysis, except for Assumption 3. To address these difficulties, we provide corresponding solutions in our analysis framework. We will also provide a better upper bound for Assumption 3 in our future work.
>
> ### **The first-order methods can not be used for the poisoning data attack application**
>
> Because attackers do not know the internal structure of the model, this application is a black box. Thus, the first-order methods can not access gradient information and are not applicable. We will clarify this in the revised version as well.
>
> [1]  Accelerated Zeroth-Order and First-Order Momentum Methods from Mini to Minimax Optimization. The Journal of Machine Learning Research, 2022

---

> ### Author Response · Authors · 2024-08-13
>
> Thank you for your time and valuable comments, especially regarding  **Assumption 3**，which is indeed strong and also used in most existing works, such as [1] and [3].  To address your concern, we further improved the variance bound and corresponding theoretical results of our algorithms.
> We first derive the following variance bound in the following inequalities (1) and (2) to replace **Assumption 3**.
> $$
> E[||\hat\nabla_x f(x,y;\zeta) - \nabla_xf_{\mu_1}(x,y)||^2] \leq \frac{2G^2}{b_1} + \frac{4d_xG^2 + \mu_1^2 \ell d_x^2}{b_1}
> \tag{1}
> $$
>  and
> $$
> E[||\hat\nabla_y f(x,y;\xi) - \nabla_yf_{\mu_2}(x,y)||^2] \leq \frac{4\ell^2D_\mathcal{Y}^2}{b_2} + \frac{8d_y\ell^2D_\mathcal{Y}^2 + 2\mu_2^2 \ell d_y^2}{b_2} \tag{2}
> $$
> Furthermore, we use the (1) and (2) instead of **Assumption 3** under our analysis framework, and adjust these parameters  $b_1 = \mathcal{O}(d_xG^2\epsilon^{-2})$, $b_2 = \mathcal{O}(d_y\ell^3\kappa D_{\mathcal{Y}}^2\epsilon^{-2})$, and by adjusting the smoothing parameters $\mu_1 = \mathcal{O}(\epsilon/(d_xL_x))$, $\mu_2 = \mathcal{O}(\sqrt{\kappa}\epsilon/(d_yL_y))$, our ZO-GDEGA still obtains the complexity of **$\mathcal{O}(\kappa^2(d_x + d_y\kappa)\epsilon^{-4})$**  for the NC-SC problem.  As for the NC-C setting, we can set $b_1 = \mathcal{O}(d_x\ell G^2)$, $b_2 = \mathcal{O}(d_y\ell D_{\mathcal{Y}}^2\epsilon^{-2})$, and by setting the smaller smoothing parameters $\mu_1 = \mathcal{O}(\epsilon/d_x)$, $\mu_2 = \mathcal{O}(\epsilon/d_y)$, our ZO-GDEGA algorithm can still obtain the complexity of $\mathcal{O}(d_x\epsilon^{-6} + d_y\epsilon^{-8})$.
>
> Thus, these derivation shows that **Assumption 3** is a strong assumption. We will add this discussion in our revised version.
>
> [1] Accelerated Zeroth-Order and First-Order Momentum Methods from Mini to Minimax Optimization. The Journal of Machine Learning Research, 2022
>
> [2] Min-Max Optimization without Gradients: Convergence and Applications to Black-Box Evasion and Poisoning Attacks.
>
> [3] Derivative-free alternating projection 490 algorithms for general nonconvex-concave minimax problems. 2023.

---

> > ### Comment · Reviewer_JCfr · 2024-08-13
> >
> > Thanks for the reply, after the rebuttal I will keep my score and recommend acceptance for this submission.

---

> > > ### Author Response · Authors · 2024-08-13
> > >
> > > We appreciate your review and valuable questions that improved our work.

---

### Author Rebuttal · Authors · 2024-08-07

We would like to thank the reviewers for the detailed reviews.  We were asked by the reviewer 4438 to provide code. Our code can be available at the link https://anonymous.4open.science/r/ZO-GDEGA-2F6E

---

### Decision · Program_Chairs · 2024-09-25

**Decision:**

Accept (poster)

**Comment:**

The paper introduces a Zeroth-Order Gradient Descent Extragradient Ascent (ZO-GDEGA) algorithm, which is designed to solve nonconvex-concave (NC-C) minimax optimization problems. The proposed algorithm has iteration complexity of  O(epsilon^-6) to find ϵ-stationary point.  The algorithm requires weaker conditions on zeroth-order gradient estimations compared to previous methods, offering a more broad/robust theoretical result. It also shows advantages in the nonconvex-strongly concave (NC-SC) case. Experimentally, the paper validates the the performance of ZO-GDEGA through numerical experiments on data poisoning attacks and AUC maximization.

The reviewers found the theoretical analysis rigorous and in line with the claims of the paper.  The reviewers also appreciate the relaxed conditions required for applying the theoretical analysis. The experiments are found in line with the message of the paper, demonstrating the relevance of the theory in practical scenarios discussed. As noted by some of the reviewers, the improvements over [43] needs further discussion. I encourage the authors to clarify this improvement further as mentioned in their response (You can use [27, Proposition 4.12] or a a clear version of it with more discussions in arxiv 2002.07919). The reviewers also noted that the paper can improve by including discussions on lower bounds and connections to first-order methods.